

# Semiautomated generation of species-specific training data from large, unlabeled acoustic datasets for deep supervised birdsong isolation

Justin Sasek[1], Brendan Allison[2], Andrea Contina[3], David Knobles[4], Preston Wilson[5] and Timothy Keitt[2]

[1] Department of Computer Science, The University of Texas at Austin, Austin, TX, United States of America
[2] Department of Integrative Biology, The University of Texas at Austin, Austin, TX, United States of America
[3] School of Integrative Biological and Chemical Sciences, The University of Texas Rio Grande Valley, Brownsville, TX, United States of America
[4] Knobles Scientific and Analysis, LLC, Austin, TX, United States of America
[5] Walker Department of Mechanical Engineering, The University of Texas at Austin, Austin, TX, United States of America

Corresponding authors
Justin Sasek, justinsasek@utexas.edu
Brendan Allison, breallis@utexas.edu

## ABSTRACT

**Background**. Bioacoustic monitoring is an effective and minimally invasive method to study wildlife ecology. However, even the state-of-the-art techniques for analyzing bird-songs decrease in accuracy in the presence of extraneous signals such as anthropogenic noise and vocalizations of non-target species. Deep supervised source separation (DSSS) algorithms have been shown to effectively separate mixtures of animal vocalizations. However, in practice, recording sites also have site-specific variations and unique background audio that need to be removed, warranting the need for site-specific data. **Methods**. Here, we test the potential of training DSSS models on site-specific bird vocalizations and background audio. We used a semiautomated workflow using deep supervised classification and statistical cleaning to label and generate a site-specific source separation dataset by mixing birdsongs and background audio segments. Then, we trained a deep supervised source separation (DSSS) model with this generated dataset. Because most data is passively-recorded and consequently noisy, the true isolated birdsongs are unavailable which makes evaluation challenging. Therefore, in addition to using traditional source separation (SS) metrics, we also show the effectiveness of our site-specific approach using metrics commonly used in ornithological analyses such as automated feature labeling and species-specific trilateration accuracy. **Results**. Our approach of training on site-specific data boosts the source-to-distortion, source-to-interference, and source-to-artifact ratios (SDR, SIR, and SAR) by 9.33 dB, 24.07 dB, and 3.60 dB respectively. We also find our approach allows for automated feature labeling with single-digit mean absolute percent error and birdsong trilateration accuracy with a mean simulated trilateration error of 2.58 m. **Conclusion**. Overall, we show that site-specific DSSS is a promising upstream solution for wildlife audio analysis tools that break down in the presence of background noise. By training on site-specific data, our method is robust to unique, site-specific interference that caused previous methods to fail.

## INTRODUCTION

An increasing number of research efforts have been focused on collecting and analyzing wildlife vocalizations in recent years (*Laiolo, 2010*; *Aide et al., 2013*; *Browning et al., 2017*; *Sugai & Llusia, 2019*; *Sugai et al., 2019*; *Teixeira, Maron & Van Rensburg, 2019*; *Desjonquères, Gifford & Linke, 2020*). Bioacoustic monitoring can be used to detect species, count the number of vocalizations per species, analyze changes in vocalization structure over time, and trilaterate the approximate location of individual animals (*Frommolt & Tauchert, 2014*; *Teixeira, Maron & Van Rensburg, 2019*; *Besson et al., 2022*). Because monitoring can be performed passively and continuously over long periods (*e.g.*, months), large datasets often accumulate, requiring automated analytical tools such as BirdNET, CARACAL, and Vesper (*Mills, 2017*; *Wijers et al., 2019*; *Kahl et al., 2021*; *Stowell, 2022*). However, the acoustic richness in these datasets can cause automated analytical approaches to perform poorly (*Evrendilek & Akcan, 2011*; *Akcan & Evrendilek, 2013*; *Zhang et al., 2014*; *Funosas et al., 2023*). For example, a signal of interest in a behavioral study may be a single faint birdsong within a complex dawn chorus, or a bird competing to be heard against road noises in an urban-rural gradient (*Priyadarshani et al., 2016*; *Wang, Venkataramani & Smaragdis, 2020*; *Dai et al., 2021*). Furthermore, each recording site is different, compounding the complexity of the problem. Here, we focus on birdsongs recorded in these noisy environments and how site-specific deep supervised source separation (DSSS) can be used to analyze them.

Any large-scale birdsong dataset analysis often begins by finding the birdsong of interest within potentially thousands of hours of audio. By running raw audio samples through a classification algorithm (*e.g.*, BirdNET; *Kahl et al., 2021*), the confidence of having recorded a bird species can be determined for a given time interval (*Acevedo et al., 2009*). Furthermore, if calibrated against known call rates, it is possible to further estimate abundances by counting the number of songs from each species (*Dawson & Efford, 2009*; *Pérez-Granados & Traba, 2021*). However, many classification algorithms are trained on mostly clean birdsongs with little interference from natural and anthropogenic sources, and their accuracy degrades with an increasing amount of competing signals (*Apol, Valentine & Proppe, 2020*). BirdNET, for instance, is trained on birdsongs from the Macaulay Library, eBird, and Xeno-Canto. In contrast to most passively collected datasets, these datasets primarily comprise clean, professionally recorded birdsongs, with only a minor presence of noisy recordings (*Planqué & Vellinga, 2005*; *Sullivan et al., 2009*; *Kahl et al., 2021*; *Funosas et al., 2023*).

Birdsong feature labeling and analysis are also informative in behavioral ecology studies (*Vehrencamp et al., 2013*). Features such as peak frequency and song amplitude can provide insights into reproductive, defense, and social behaviors (*Brumm & Todt, 2004*; *Teixeira, Maron & Van Rensburg, 2019*). These insights allow us to measure how different

environmental changes and anthropogenic activities impact bird behavior. For example, an upward shift in song frequency has been observed in several species in response to anthropogenic noise (*Halfwerk & Slabbekoorn, 2009*; *Cardoso & Atwell, 2012*; *Nemeth et al., 2013*).  In many cases, birdsong feature labeling and analysis have focused on the minimum and maximum frequency of songs. The labeling process often involves manually annotating the spectrograms because background noise makes automating this process challenging. Even then, some human-labeled error is as high as 300–1,800 Hz (*Brumm et al., 2017*). It has been argued that the subjective process of identifying minimum frequencies on a spectrogram would be better served by an objective, power-based approach (*Cardoso & Atwell, 2012*). Yet, these approaches typically require clean audio. Any interference that exceeds the joule-per-second power threshold could be mislabeled as the minimum or maximum frequency.

Another application that is negatively impacted by background noise is trilateration, where the source of a sound is spatially located. By spacing out several microphones, the time difference of arrival between microphones can be used to trilaterate the location of an animal (*Grande et al., 2019*). This can be used to autonomously track bird movement to and from their nest (*Nathan et al., 2022*). Or, if obtaining population counts, trilateration can be used to prevent birds from being counted twice. However, with any background noise, finding the time difference of arrival becomes increasingly hard (*Evrendilek & Akcan, 2011*; *Akcan & Evrendilek, 2013*).

Due to interference during recording, the original ground-truth birdsong is often lost, making it challenging to objectively define a clean birdsong. Consequently, we loosely define a clean birdsong based on the degradation's impact on the specific bioacoustics application. For birdsong species classification tasks, a clean birdsong can be confidently labeled by an expert ornithologist, while a noisy birdsong is challenging to identify due to interference. Similarly, for birdsong feature analysis, clean birdsongs can be confidently labeled by an ornithologist, whereas noisy birdsongs may pose difficulties. In trilateration applications, the noisiness of a birdsong can be measured by the resulting error in the estimated position.

One upstream solution to these issues is to develop robust methodologies for extracting clean birdsongs from audio recordings. However, this is challenging for several reasons. At first glance, one may consider using visual semantic separation. This technique assigns each pixel from an image to a source, often using a machine learning model. For instance, for an autonomous car, pixels from the road lines would be assigned accordingly (*Rahman, Khan & Barnes, 2020*; *Ren et al., 2023*). Audio waveforms can be converted into spectrograms using Fourier transformations and then each time-frequency bin could be treated as a pixel. However, unlike pictures, audio is additive: each time-frequency pair could have multiple sources. For instance, if a certain frequency is heard during a concert, it could have come from several instruments at the same time. Furthermore, some bird vocalizations can be very similar, making it even harder to differentiate between them.

Some birdsong source separation (SS) approaches use unsupervised machine learning which learns from unlabeled datasets where there is no ground truth. For example, (*Sun, Yen & Lin, 2022*) use non-negative matrix factorization (NMF) to separate audio samples

into similar categories. For instance, it may group low road noises into one category and higher human speech into another (*Sun, Yen & Lin, 2022*). However, NMF is limited in complexity because it is purely a linear method. A neural network is preferred because it can better represent more complex function spaces such as birdsong SS (*Thakur & Konde, 2021*). This is what *Denton, Wisdom & Hershey, 2021a*; *Denton, Wisdom & Hershey, 2021b* do with Bird MixIT, a bird SS tool that uses deep unsupervised mixture invariant training (MixIT). By learning to separate mixtures of noisy birdsongs from one another, the model learns to separate individual birdsongs from the original noisy recordings (*Wisdom et al., 2020*; *Denton, Wisdom & Hershey, 2021a*).

However, this approach lacks control and is less customizable. Besides constructing the training dataset, there is little one can do to control the final model. These models often learn to separate the audio in a random order, splitting some sources and combining others. For instance, looking at example 4 from Bird MixIT's Caples sample outputs, the Rebut class is combined with the background noise using the first model, and the Btywar class is combined with the background noise using the second (*Denton, Wisdom & Hershey, 2021b*). Because the training process is unsupervised, it is very challenging to address this or any other undesirable behaviors.

In contrast, supervised learning turns the SS into a training dataset creation problem. Because supervised learning models learn to replicate the training dataset, a high-quality training dataset often means a high-quality model. This offers greater control and is more customizable. One can specify exactly what sounds constitute the foreground and the background, and the model will learn to separate them accordingly. For instance, if trying to separate the Rebut and Btywar birdsong classes from background noise, each could be assigned a specific output channel of a supervised SS model, and there would be no ambiguity in the model's desired behavior.

Fortunately, several deep supervised audio SS models exist such as ConvTasNet (Convolutional Time-Domain Audio Separation Network), DPRNNTasNet (Dual-path Recurrent Neural Network Time-Domain Audio Separation Network), and SuDORMRFNet (Successive Downsampling and Resampling of Multi-Resolution Features Network) where ConvTasNet and SuDORMRFNet use convolutional neural networks (CNN) and DPRNNTasNet uses a recurrent neural network (RNN) (*Luo & Mesgarani, 2019*; *Luo, Chen & Yoshioka, 2020*; *Tzinis, Wang & Smaragdis, 2020*). While these and several other supervised SS models exist, the overwhelming majority of these models require clean signals from each source as the ground truth for training. This is seen in the music SS dataset MUSDB18 where each instrument is recorded separately and without interference (*Rafii et al., 2017*). However, for birdsong SS, obtaining a clean signal of both the birdsongs and the background is challenging because the recording does not occur in a controlled environment. (*Priyadarshani et al., 2016*; *Wang, Venkataramani & Smaragdis, 2020*; *Dai et al., 2021*). And, if SS models are trained on audio with even a little interference, the final output will have that same interference (*Lones, 2023*).

Conveniently, several publicly-available birdsong datasets exist with cleanly-recorded birdsongs, and some approaches have trained deep supervised SS models on these datasets. By mixing together different species of animals whose vocalizations interfere, these

approaches were successfully able to separate the different vocalizations (*Izadi, Stevenson & Kloepper, 2020*; *Bermant, 2021*; *Zhang et al., 2022*). However, due to site-specific variations in birdsongs and background noise, these approaches' practicality for actual field use is questionable. We show that ideally, each source separation model should be fine-tuned for the recording sites of interest to account for this variation.

Here, to the best of our knowledge, we are the first to train a deep supervised source separation (DSSS) model using site-specific, passively-recorded data. We introduce a site-specific DSSS workflow that addresses the challenges associated with supervised SS with a two-step process. First, we create a site-specific synthetic dataset by extracting, denoising, and mixing sufficiently clean birdsongs and background noise from passive acoustic datasets. Second, we train on the dataset. Unlike previous DSSS methods, our model learns to remove background noise specific to that site, and compared to unsupervised SS, our approach has more control and is more customizable.

We then test our methodology using traditional SS metrics such as source-to-distortion ratio, source-to-interference ratio, and source-to-artifact ratio (SDR, SIR, and SAR) as well as bioacoustics-inspired metrics such as automated feature labeling accuracy and species-specific trilateration accuracy. These additional metrics both highlight potential downstream SS applications and more rigorously vet our model.

## MATERIALS & METHODS

We present a semiautomated workflow to train a site-specific DSSS model using raw field recordings to isolate Golden-Cheeked Warbler (GCW) birdsongs. The Golden-Cheeked Warbler (*Setophaga chrysoparia*) is a federally endangered migratory songbird that breeds only in mature Ashe juniper (*Juniperus ashei*) and oak (*Quercus* spp.) woodlands within central Texas. This species vocalizes three types of songs: A, B, and C; however, C songs do not occur often, and we do not include them in our study (*Groce et al., 2010*). Our analysis proceeded as follows:

### Obtain audio from recording sites

Passive acoustic recording was conducted by David Knobles (Knobles Scientific and Analysis, LLC) and Preston S. Wilson (Walker Department of Mechanical Engineering) using SM4 recorders placed in remote locations on the Balcones Canyonlands Preserve. Field permits were provided by the City of Austin Wildland Conservation Division with the approval numbers Knobles-2016, Knobles-2017, Knobles-2018, and Knobles-2019. Besides installation and servicing, there was no human involvement. For our training dataset, we obtained 1,425 six-second birdsongs from 13 unique recording sites deployed between 2016 and 2018. This entailed the breeding seasons at two, seven, and nine locations in 2016, 2017, and 2018. For our testing dataset, we obtained 1,449 six-second birdsongs from three additional recording sites in 2019. Overall, because GCWs are territorial, we assume that these recordings correspond to between 16 and 21 CGWs depending on whether birds returned the following year (*Bolsinger, 2000*).

## Obtain clean birdsong audio

Next, we sorted through the audio recordings to find clean GCW birdsongs. We used BirdNET to scan the recordings with a sliding three-second window incremented by one second. We similarly ran Pretrained Audio Neural Networks (PANNs), a general audio classification deep learning model, over our recordings to capture less common events such as ambulances, music, or construction (*Kong et al., 2020*). Next, we found the joint probability of each three-second window both containing a GCW and not containing any other category. From this list, we selected the top 1500 windows and extracted these audio segments with a 1.5-second buffer before and after to ensure the entire birdsong was captured.

Then, we performed high-pass Butterworth filtering to remove all noise lower than the birdsong. We then normalized the peak frequency to −1 dB and performed stationary spectral gating noise reduction. While some noise remained, it was outside the human audible spectrum (20 Hz to 20 kHz), deeming the outputted birdsong mostly clean. At this point, we obtained GCW audio samples with the top community ratings from the Macaulay Library and also cleaned this audio using the same process. This further diversified our dataset.

After generating spectrograms, we manually sorted through the samples for clean birdsongs: we first removed all recordings where there was interference from other bird species. We then sorted the recordings by their birdsong-background ratio. Any recordings with the birdsong more than 10 dB quieter than the background noise were removed. This yielded 529 clean birdsongs. We also manually sorted the samples into A song, B song, and any other unique variations. Aside from the actual data collection itself, this sorting is the only manual aspect of our workflow.

For the testing set, we repeated this process, yielding 218 clean birdsongs.

## Obtain background audio

To obtain background audio, we removed the segments marked positive for GCW birdsong by BirdNET from 300 h of randomly selected raw audio from the recording sites. We then condensed each hour to its loudest five minutes: we first took a one-second rolling summation over the absolute value of each waveform value to smooth out the audio. We then took a one-second rolling maximum to ensure that the audio was not fragmented into segments smaller than two seconds. Finally, we extracted the top five minutes of audio using this calculated metric. We find this extraction technique removes most of the plain white noise from each hour and leaves the most interesting audio, such as cars passing or people talking. We also pulled the top audio segments from PANNs' unique classification categories to diversify the dataset (*Kong et al., 2020*). For each category, we found the joint probability of each three-second window both containing that category and not containing a GCW birdsong. Finally, we used BirdNET's results to find the 10 most common birds in our recordings (excluding the GCW) and obtained recordings of those species from the Macaulay Library to further diversify our background dataset.

We repeat this process for the testing recording sites, excluding the recordings from the Macaulay Library, as those are only used to enhance our training dataset. Overall, this yielded 33 h of background audio for training and 29 for testing.

### Create data generator

Our data generator randomly stitches background samples together and then randomly layers GCW birdsongs on top. To maintain a realistic background with consistent volume, we chose not to layer multiple background recordings. To ensure a balanced dataset, we increased the probability of uncommon GCW song types such as A songs with no trailing hook. To increase dataset diversity, birdsongs were randomly time-stretched by −20% to 20% and amplified by −85% to 20%. We also pitch-shifted the birdsongs to reflect the birdsong peak frequency variance in our dataset. For the background, we reduced the volume by 90% 5% of the time and by 95% 5% of the time. This ensured that the model also worked with little background noise.

### Train model

To train our model, we used Asteroid with a three-second window and negative pairwise SDR for the loss function. Asteroid is an audio research tool that allows the training of several deep supervised source separation (DSSS) models on the same dataset (*Pariente et al., 2020*). We trained on ConvTasNet, DPRNNTasNet, and SuDORMRFNet (*Luo & Mesgarani, 2019*; *Luo, Chen & Yoshioka, 2020*; *Tzinis, Wang & Smaragdis, 2020*). These models were selected for being released in or after 2019, being compatible with Asteroid, and obtaining an SDR above 7.1 dB on the ESC50 dataset (*Piczak, 2015*). To measure the importance of using site-specific audio, we retrain SuDORMRFNet (the highest performing architecture) without any site-specific audio, separating the Macaulay Library Golden-Cheeked Warbler birdsongs from the others.

### Evaluate model

Next, we ran 10 automated evaluation tests on our model. The first five tests measured the output SDR, SIR, and SAR after different input adjustments: song type, birdsongs per minute, background type, birdsong-background ratio, and random forest impulse response type. For song type, we used audio only including either A song, A song with a hook, A song with no hook, or B song. For birdsongs per minute, we varied the birdsongs per minute from 0 to 20. To test background type, we layered two unique backgrounds on top of each other to verify that the model could handle unseen environments and separate birdsongs from multiple sources of interference at once. We also investigated different birdsong volume ratios which we calculated by dividing the total volume of the birdsongs over that of the background audio. For our test, we used 20 birdsongs over 60 s of audio with a ratio of either 0.01, 0.02, 0.05, 0.1, 0.2, 0.5, 1, 1.5, or 2. For reference, birdsongs with ratios of 0.01 and 0.02 are barely audible to the human ear. Our final robustness test used random forest impulse responses which capture how sound travels through a randomly generated, virtual forest. We generated these impulse responses for virtual forests with 100,000, 200,000, 500,000, and 1,000,000 trees using Microsoft's virtual forest impulse response generator. Birdsongs were convolved with these virtual forest impulse responses

and used as the ground truth to simulate distortion from different-sized forests. For each robustness test, we used between 4.20 and 7.17 h of synthetic data as the ground truth. To better emulate field recordings, we used one-minute audio samples; however, our model input size is three seconds, so we split the longer samples into three-second chunks with a one-second overlap and merged the outputs with a half-second crossfade.

We also visually inspected several birdsongs that we set aside for testing. These tests do not offer objective performance measures but still provide visual insight into our model. We first ran our model on several noisy birdsongs and visually inspected the results. Additionally, using previous birdsong feature annotations, we set aside several outlier birdsongs that constituted the extreme values, both the lowest and highest, of the feature labels. For A songs, these features included the number of repeat elements, main buzz duration, and peak frequency. For B songs, the features were the initial buzz duration, presence of a short note after the initial buzz, main buzz duration, peak frequency excluding flourish, and presence of flourish after the birdsong. Then, we ran our model and visually inspected the results to ensure our model could handle outlier birdsongs.

We also tested for potential in downstream applications, starting with automated feature labeling. We created a basic proof-of-concept for automated feature labeling which automatically measures the minimum and maximum birdsong frequencies. First, we trimmed each sample by taking each six-second birdsong waveform, squaring each value, and smoothing out any noise by finding the moving average with a sliding six-second window. We then selected the largest value from the moving average to find the center point of the birdsong and the surrounding 1.5 s from the center point to obtain a three-second sample. Then, we converted each sample into a spectrogram and leveraged a Canny filter to identify the highest and lowest edges. This gave us the minimum and maximum frequencies. After generating 720 six-second birdsong samples, we used this labeling process on the true isolated birdsongs to estimate the ground truth. Then, we added background audio, ran our labeling process, ran our model, and then did labeling again. By measuring the percent absolute error before and after running our model, we evaluate the effectiveness of our model.

We used a similar method to test timing preservation. By finding the maximum cross-correlation before and after running our model, we were able to assert that there was no time difference introduced by our model over 1,000 trials. If our model shifted the input audio by even a few milliseconds, it would introduce large errors during trilateration. Knowing this was not the case, we created a basic proof-of-concept for species-specific trilateration. We coded a simulated trilateration environment in a forest with microphones placed 1.5 m above the ground—one at the origin, one 10 m north, and one 10 m east. For all 918 trials, we randomly picked a location for the bird within 10 m of at least one microphone. We again used Microsoft's virtual forest impulse response generator to create impulse responses that capture the distortion of the forest as well as the time difference of arrival between the bird and the microphones. These impulse responses were then applied to the birdsong using a convolution. We then layered unmodified background audio on top of the birdsong to simulate trilateration interference. After running the combined audio through our DSSS model, we found the maximum cross-correlation between each

microphone input which gave us the time difference of arrival between the bird and each microphone. With these time differences, we trilaterated each bird's location by solving a system of time difference of arrival equations.

We also evaluated our model's effect on BirdNET (*Kahl et al., 2021*). We ran our model on 2401 six-second audio segments with a GCW birdsong and 2401 without for each of the above birdsong-background ratios. We then found the ideal classification probability threshold before and after separation to find the maximum possible accuracy.

## RESULTS

We found that the SuDORMRFNet model yielded the highest SDR, SIR, and SAR after seeing 250,000 six-second samples. Training ConvTasNet, another convolutional model, also yielded a high SDR, SIR, and SAR suggesting that convolutional neural networks outperform recurrent neural networks in this domain.

After retraining SuDORMRFNet without site-specific audio, the SDR, SIR, and SAR decrease by 9.33 dB, 24.07 dB, and 3.60 dB respectively, highlighting the importance of site-specific audio in source separation. Notably, the performance drop in the source-to-interference ratio (SIR) suggests that this model was unable to handle background interference specific to this site (Fig. 1A).

### Model Robustness

Our five robustness tests all obtained an SDR, SIR, and SAR above 19 dB. For reference, the leading model for the music SS dataset MUSDB18 only obtained an SDR of 9.200 dB (*Rafii et al., 2017*; *Rouard, Massa & Défossez, 2022*). Because we used site-specific data and a robust data generator, we were able to maintain high SS metrics across the entire training distribution. This means our model was able to interpolate well from examples it saw during training. Notably, we had very few training samples for A songs with no trailing hook; however, by inflating its occurrence in our data generator, we maintained the same model performance as other song types (Fig. 1B). Our birdsong occurrence frequency test was also inside the training distribution. Even if there are 20 birdsongs in a minute, with some potentially overlapping each other, the model obtained metrics above 22 dB (Fig. 1C). The same was true for our birdsong-background ratio test: for all recordings, we obtained metrics above 19 dB even though recordings with ratios at or below 0.02 are barely perceivable to the human ear (Fig. 1D).

Outside of the training distribution, our model also achieved SS metrics above 27 dB, including for different background and virtual forest types. During training, we only used one unique background recording at a time with no mixing; however, for one test, we mixed two unique backgrounds on top of one another. Despite this, the results remained relatively unphased with the extra background noise (Fig. 1E). We also convolved the isolated birdsong with random forest impulse responses. Despite not having trained with this functionality, our model still obtained SS metrics above 27 dB (Fig. 1F). The model learned to preserve this convoluted distortion solely from the dataset's diversity and data augmentation. Looking closer, we visually inspected isolated birdsongs with and without convolution, and we saw that the convolution distortion is maintained (Figs. 2C and 2D).

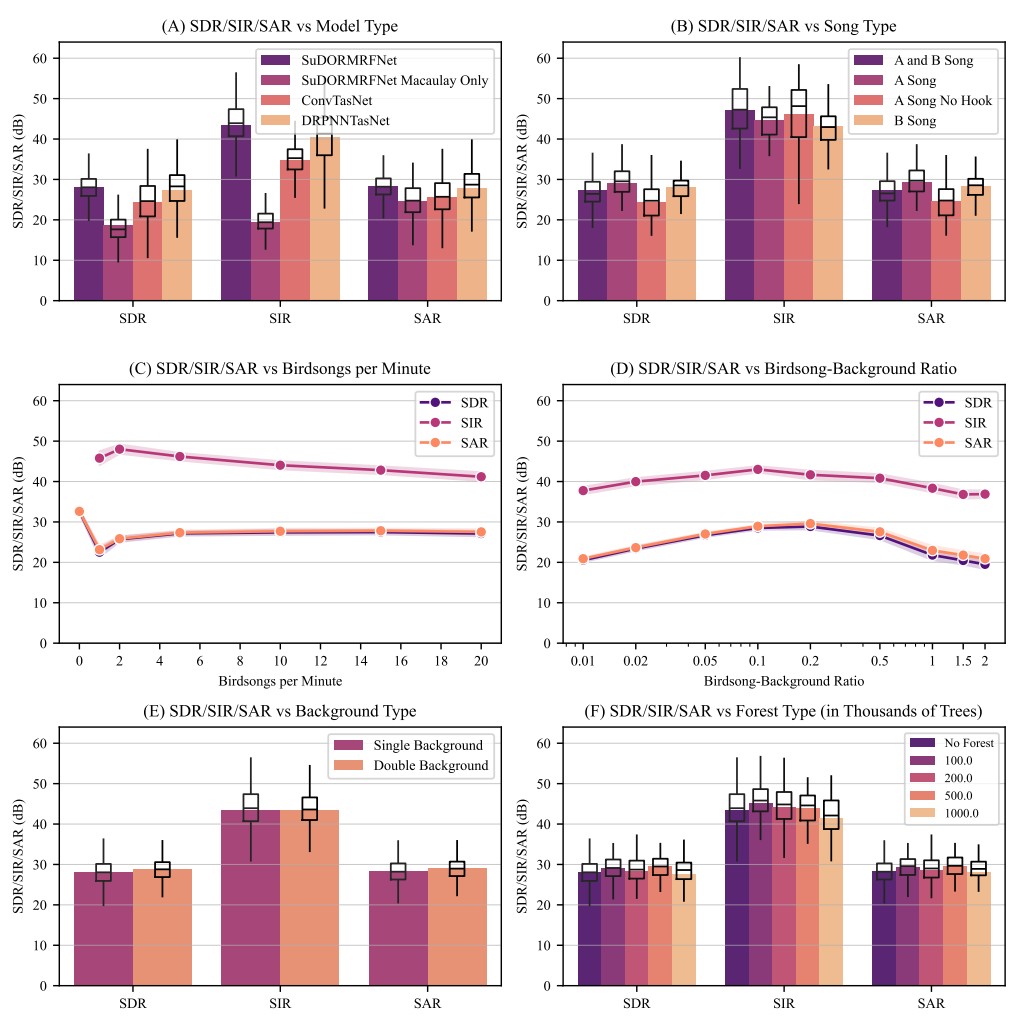

**Figure 1** **Model performance for different model types, song types, birdsong counts per minute, birdsong-background ratios, background types, and random forest impulse response types.** In these figures, we evaluated our deep supervised source separation (DSSS) model by comparing true isolated birdsongs with our model's estimations after adding background audio and performing source separation. We evaluated with source-to-distortion source-to-interference, and source-to-artifact ratios (SDR, SIR, and SAR) where higher values are better. For reference, the leading model for the music SS dataset MUSDB18 obtained an SDR of 9.200 dB (*Rafii et al., 2017*; *Rouard, Massa & Défossez, 2022*). (A) SuDORMRFNet performed the best with an SDR of 28.01 dB, SIR of 43.53 dB, and SAR of 28.25 dB after evaluating all of our models on between 4.20 and 7.17 h of synthetic data. Overall, we see convolutional approaches outperform recurrent ones. Additionally, we also trained a model only on recordings from the Macaulay Library with other species of birds constituting the background audio. This performs substantially worse with the SDR, SIR, and SAR decreasing by 9.33 dB, 24.07 dB, and 3.60 dB respectively. This shows the importance of training with site-specific recordings. (B) Our model obtained an SDR, SIR, and SAR above 24 dB for all song types. (C) Our model obtained an SDR, SIR, and SAR above 22 dB regardless of the birdsong occurrence frequency. (D) For all birdsong-background ratios, we obtained an SDR, SIR, and SAR above 19 dB. For reference, birdsongs with ratios of 0.01 and 0.02 are barely audible to the human ear. 

**Figure 1 (...continued)**
(E) After doubling the background, we saw no reduction in the SDR, SIR, or SAR. (F) Our model obtained an SDR, SIR, and SAR above 27 dB in the presence of random forest impulse responses. For panels A, B, E, and F, we overlay boxplots, and for panels C and D, we show 90% confidence intervals.

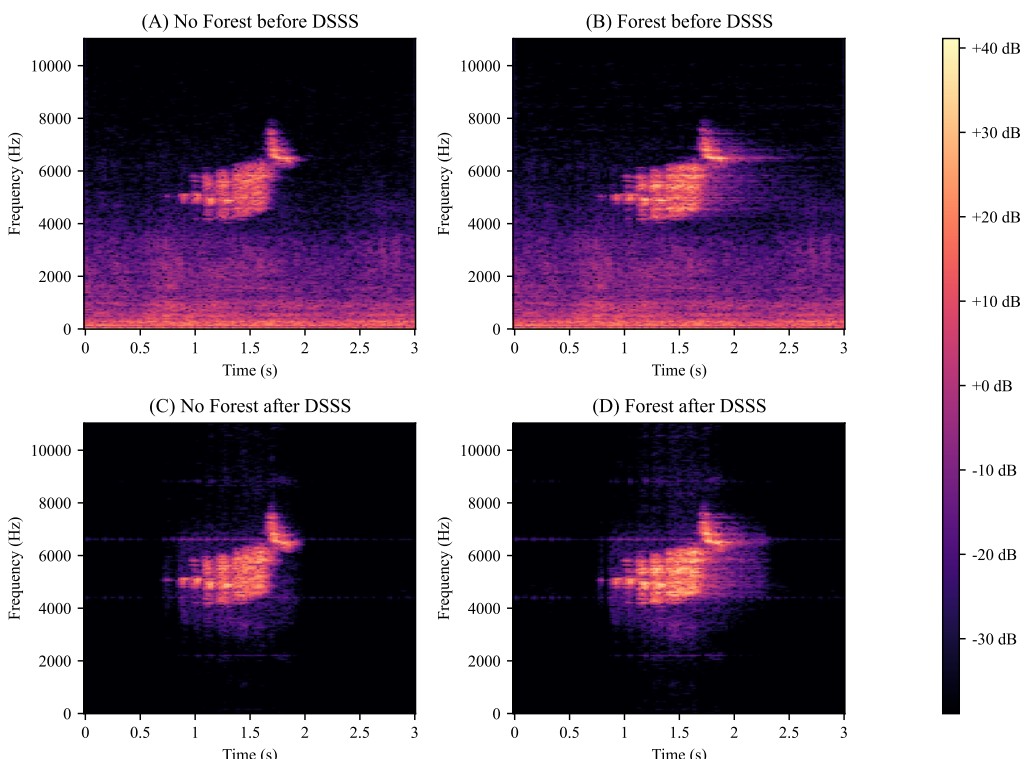

**Figure 2** **Random forest impulse response distortion preservation.** (A) Golden-Cheeked Warbler (GCW) recording mixed with background audio. (B) Same recording from panel A except the birdsong is convolved with a 1,000-thousand-tree virtual forest impulse response. This simulates the audio traveling through a forest and bouncing between the trees. (C) The recording from panel A ran through our deep supervised source separation (DSSS) model. (D) The recording from panel B ran through our deep supervised source separation (DSSS) model. Note how the convolution distortion is maintained by our model even though no convolutions were used in training. This shows that our model did not overfit on the training data and is robust to birdsong feature variations.

## Visual inspection

We then ran our model on several noisy birdsongs that we reserved for testing. The resulting spectrograms suggest that our model removes most of the background noise (Fig. 3). We also ran our model on several outlier birdsongs, and our model appeared to have preserved all features (Fig. 4).

## Automated feature labeling

Next, we made an automated feature labeling simulation. Without the model, our algorithm labels the minimum and maximum frequencies with mean absolute percentage errors of 58.85% and 29.32% respectively. With our model, this improves to 1.62% and 0.77%

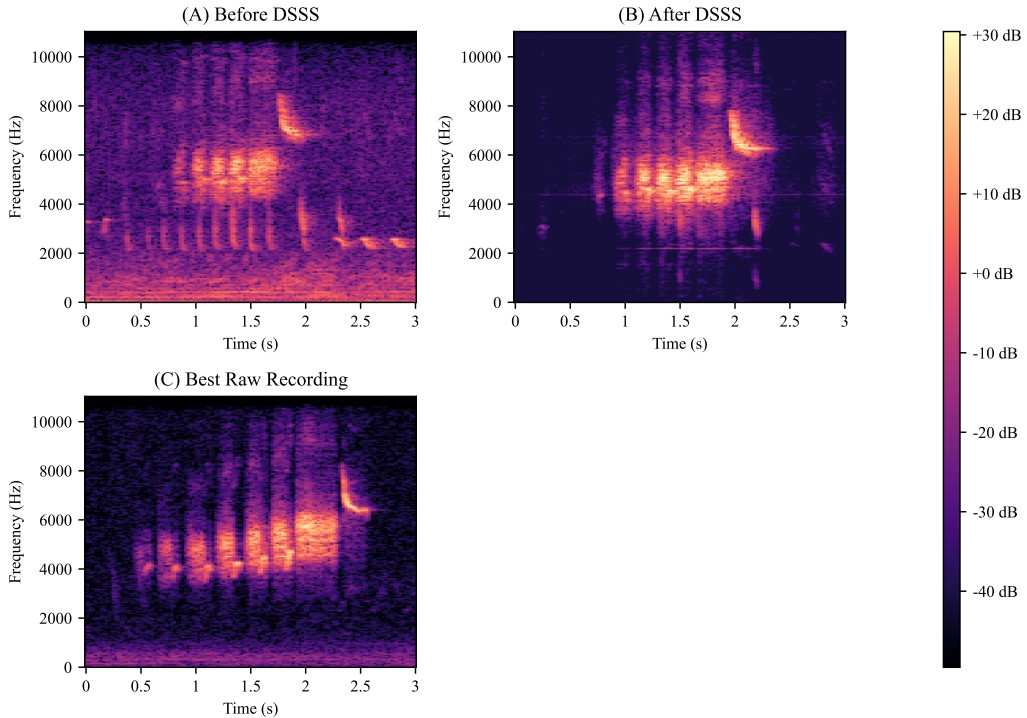

**Figure 3** **Deep supervised source separation on noisy birdsong.** (A) Golden-Cheeked Warbler (GCW) birdsong recording with interference from road noise and other species of birds. (B) The same recording after passing through our deep supervised source separation (DSSS) model. Note the strong reduction in interference. (C) GCW birdsong recording that was provided by the Macaulay Library and that was the cleanest raw recording in our dataset. Note that this recording has similar interference as panel B.

respectively. Notably, our model is most helpful for low birdsong-background ratios. For example, samples with background audio that is 100 times louder than the birdsong have a mean absolute percent error above 96% before running our model and a single-digit error after (Table 1).

## Timing preservation and trilateration simulation

Finally, we tested birdsong timing preservation by finding the time difference between the clean input signal and the clean signal mixed with background audio and then ran through DSSS. Out of 1,000 trials, all are aligned down to 45.4 µs, the best possible given a 22,050 Hz sampling rate.

With timing preserved, we ran a trilateration simulation with a birdsong coming from one source and the background audio coming from another. Without any SS, the mean error is 9.3 m with only 2.57% of the birdsongs being trilaterated within 1 m. We found that our model significantly improves these results with a mean error of 2.58 m and 58.86% of the birdsongs being trilaterated within 1 m (Table 2).

## BirdNET improvement

Our model has no statistically significant impact on BirdNET except for birdsong-background ratios of 0.2 where a slight improvement is seen (Table 3). However,

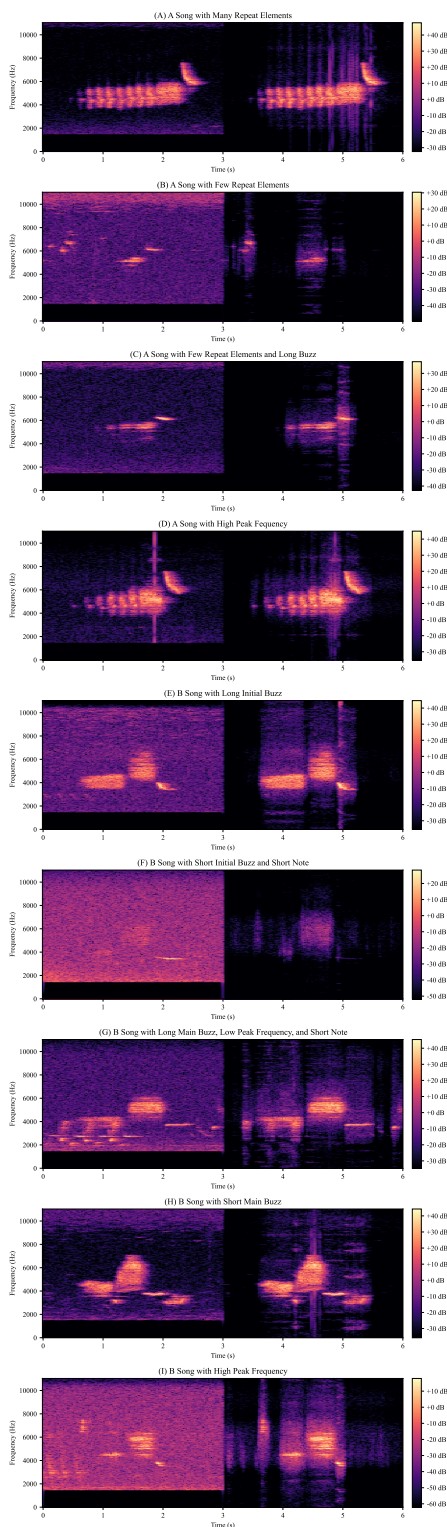

**Figure 4 Deep supervised source separation on outlier birdsongs.** Using previous Golden-Cheeked Warbler (GCW) feature annotations, we set aside several (continued on next page...)

**Figure 4 (…continued)**
outlier birdsongs for evaluation. For instance, an outlier A song may have many repeat elements as seen in panel A or very few as seen in panel B. For each panel, we give a short description of what made that birdsong an outlier, show the original recording on the left, and show the recording after deep supervised source separation (DSSS) on the right. This figure shows that our model appears to preserve birdsong features, suggesting that automated feature labeling is possible.

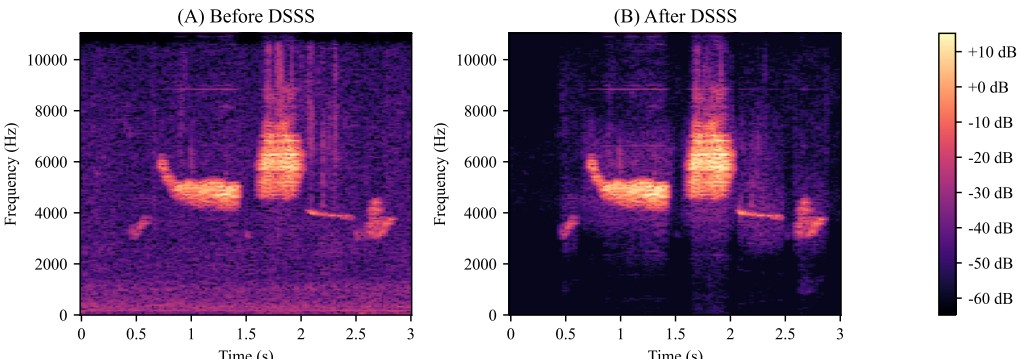

**Figure 5** **BirdNET misclassification after source separation.** (A) Golden-Cheeked Warbler (GCW) B song. (B) The recording from panel A ran through our deep supervised source separation (DSSS) model. While BirdNET correctly identified panel A as a GCW birdsong with 1.91% confidence, BirdNET did not detect the GCW birdsong in panel B due to the 1% detection threshold, even though there is less noise. This highlights the fragility of BirdNET and suggests why our model did not improve BirdNET's performance.

after further investigation, we found that BirdNET—in addition to too much noise—underperforms when there is too little noise. After our model removes much of the background interference in Fig. 5, the classification confidence drops from 1.91% to below 1%, the detection threshold.

## DISCUSSION AND CONCLUSIONS

Overall, we implemented a site-specific deep supervised source separation (DSSS) approach to isolate birdsongs from field recordings containing interference from humans and other bird species. Across our five robustness tests, our model obtained consistently high SDRs, SIRs, and SARs even with faint birdsongs and different background noise levels.

However, while SDR, SIR, and SAR are commonly used metrics for SS, they have shortcomings in bioacoustics. For example, a model could have a high SDR, SIR, and SAR, but distort the minimum and maximum frequencies. Because these signal boundaries are only a fraction of the SDR, SIR, and SAR metrics, they could easily be overlooked. This is problematic because distorted signal boundaries could drastically alter research results, such as in the numerous studies observing an upward shift in song frequency in response to anthropogenic noise (*Halfwerk & Slabbekoorn, 2009*; *Cardoso & Atwell, 2012*; *Nemeth et al., 2013*).

Consequently, we also tested our model specifically for downstream bioacoustics applications. Visual inspection confirmed unique birdsong variations like high maximum

**Table 1 Automated feature labeling performance for minimum and maximum frequencies.** By leveraging a Canny filter, we estimated the ground truth minimum and maximum frequencies of the clean birdsongs in our testing dataset. Then, we layered background audio on top and re-measured the minimum and maximum frequencies before and after being run through our model. This background audio caused significant error which our DSSS model was able to reduce down to single-digit absolute percent error. Notably, our model is most helpful for low birdsong-background ratios when the background is louder than the birdsong. For all numbers, 90% confidence intervals are shown.

| Birdsong-Background Ratio | Min frequency absolute percent error | | Max frequency absolute percent error | |
|---|---|---|---|---|
| | Before | After | Before | After |
| 0.01 | 97.49 ± 1.32% | 4.31 ± 1.47% | 96.07 ± 1.74% | 1.01 ± 0.38% |
| 0.02 | 98.5 ± 1.67% | 4.04 ± 1.69% | 89.09 ± 4.22% | 2.03 ± 1.21% |
| 0.05 | 98.53 ± 1.36% | 0.83 ± 0.52% | 43.47 ± 8.14% | 0.51 ± 0.44% |
| 0.1 | 100.01 ± 1% | 0.84 ± 0.68% | 23.69 ± 6.49% | 0.59 ± 0.43% |
| 0.2 | 93.54 ± 4.28% | 0.42 ± 0.19% | 9.19 ± 4.27% | 0.54 ± 0.64% |
| 0.5 | 34.82 ± 8.57% | 1.02 ± 0.88% | 1.13 ± 0.71% | 0.49 ± 0.45% |
| 1 | 4.47 ± 3.49% | 1.55 ± 1.2% | 0.6 ± 0.16% | 0.65 ± 0.56% |
| 1.5 | 1.91 ± 2.08% | 1.28 ± 0.96% | 0.49 ± 0.21% | 0.92 ± 0.84% |
| 2 | 0.37 ± 0.25% | 0.32 ± 0.19% | 0.16 ± 0.06% | 0.17 ± 0.09% |
| Average | 58.85 ± 2.97% | 1.62 ± 0.34% | 29.32 ± 2.61% | 0.77 ± 0.21% |

**Table 2 Automated trilateration performance before and after deep supervised source separation.** We simulated birdsong trilateration in a forest with microphones placed 1.5 m above the ground—one at the origin, one 10 m north, and one 10 meters east. Then, we layered in background audio at the origin. Before running our model, we were able to pinpoint the location of the birdsong with a mean absolute error of 9.3 m with 2.57% of all samples being within 1 m. After running our model, we achieved mean absolute error of 2.58 m with 58.86% of all samples being within 1 m. This suggests that deep supervised source separation (DSSS) is a feasible upstream solution to trilateration. For all numbers, 90% confidence intervals are shown.

| Birdsong-Background Ratio | Absolute error | | Samples within 1 m | |
|---|---|---|---|---|
| | Before | After | Before | After |
| 0.01 | 9.25 ± 0.71 m | 2.9 ± 0.75 m | 0 ± 0% | 55.26 ± 9.44% |
| 0.02 | 9.31 ± 0.79 m | 3.14 ± 0.82 m | 1.32 ± 2.16% | 52.63 ± 9.48% |
| 0.05 | 9.53 ± 0.77 m | 2.82 ± 0.84 m | 1.25 ± 2.06% | 58.75 ± 9.11% |
| 0.1 | 9.43 ± 0.79 m | 2.45 ± 0.76 m | 1.32 ± 2.16% | 59.21 ± 9.33% |
| 0.2 | 9.53 ± 0.77 m | 2.31 ± 0.73 m | 1.25 ± 2.06% | 61.25 ± 9.02% |
| 0.5 | 9.41 ± 0.79 m | 2.4 ± 0.77 m | 1.32 ± 2.16% | 60.53 ± 9.28% |
| 1 | 9.37 ± 0.8 m | 2.41 ± 0.74 m | 2.5 ± 2.89% | 60 ± 9.07% |
| 1.5 | 9 ± 0.88 m | 2.4 ± 0.77 m | 6.58 ± 4.71% | 60.53 ± 9.28% |
| 2 | 8.91 ± 0.88 m | 2.41 ± 0.74 m | 7.5 ± 4.87% | 61.25 ± 9.02% |
| Grand Total | 9.3 ± 0.27 m | 2.58 ± 0.26 m | 2.57 ± 0.98% | 58.86 ± 3.06% |

frequencies and distortions from environmental effects like forest reverberations were accurately captured. This suggests our model learned the nuances of different birdsongs rather than overfitting the training data. Preserving these distinctive features enables

**Table 3 BirdNet classification improvement vs birdsong-background ratio.** Here, we test how well BirdNet can determine if a recording has a Golden-Cheeked Warbler vocalization or not. We find that preprocessing audio with our model makes no statistically significant difference in the classification accuracy except for birdsong-background ratios of 0.2 where a slight improvement is seen. However, we hypothesize this is caused by the fragility of BirdNET. For some samples, removing noise made the classification probability worse, as seen in Fig. 4. For all numbers, 90% confidence intervals are shown.

| Birdsong-Background Ratio | Accuracy Before | Accuracy After |
|---|---|---|
| 0.01 | 92.88 ± 2.59% | 91.39 ± 2.83% |
| 0.02 | 89.51 ± 3.09% | 91.76 ± 2.77% |
| 0.05 | 94.76 ± 2.25% | 96.63 ± 1.82% |
| 0.1 | 93.26 ± 2.53% | 94.38 ± 2.32% |
| 0.2 | 91.76 ± 2.77% | 95.88 ± 2% |
| 0.5 | 92.88 ± 2.59% | 92.51 ± 2.65% |
| 1 | 90.64 ± 2.94% | 91.01 ± 2.88% |
| 1.5 | 87.59 ± 3.33% | 87.97 ± 3.29% |
| 2 | 87.22 ± 3.37% | 87.59 ± 3.33% |
| Average | 91.17 ± 0.95% | 92.13 ± 0.9% |

automating sophisticated analyses like peak frequency and buzz length measurement. Currently, birdsong labeling analysis is often manual because background noise makes automating this process very challenging. However, if automated with source separation (SS), this would allow large-scale feature analysis without human bias. By using our DSSS model, we were able to algorithmically extract minimum and maximum birdsong frequencies with single-digit mean percent error. While not tested, we predict other features such as buzz duration could be similarly extracted.

The current approaches for trilateration also break down with background noise (*Evrendilek & Akcan, 2011*; *Akcan & Evrendilek, 2013*; *Zhang et al., 2014*), and our DSSS workflow is again a potential solution. By isolating a single species of bird, the time differences of arrival can be extracted using cross-correlation. Then, the location of each birdsong can be calculated with a mean absolute error of 2.58 m.

Finally, while our DSSS model did not improve BirdNET's performance, we hypothesize this is caused by BirdNET's lack of robustness. As seen in Fig. 5, our model eliminated a lot of the background noise, but surprisingly, BirdNET's performance decreased. In addition to very noisy recordings, we hypothesize that BirdNET was not trained on such very clean recordings and consequently failed during our tests. While our model likely helps in some cases, this benefit is canceled out by BirdNET's fragility with distribution shifts. Looking forward, other classification models may work better. *Denton, Wisdom & Hershey (2021a)* and *Denton, Wisdom & Hershey (2021b)* saw improvement in their custom classification model after running their source separation model. However, they used unsupervised learning without site-specific audio. Or, SS could instead be used to augment the classifier's training set. A SS model could be used to isolate birdsongs for various species. Then, the isolated birdsongs could be mixed with diverse background audio

to create a more robust dataset and classification model. Unlike BirdNET, this classifier would be robust to recordings with very little noise if some isolated birdsongs were left unmixed.

Overall, this research builds on previous methods because it is more customizable and has improved site-specific performance. Because our approach is supervised, the inputs and outputs are fully controllable. And, by training on site-specific recordings, we boost site-specific performance by 9.33 dB, 24.07 dB, and 3.60 dB for SDR, SIR, and SAR respectively.

While testing was limited to Golden-Cheeked Warblers, we expect our approach to be easily adaptable to all species of birds. Likewise, while only testing a few downstream applications, our approach should be adaptable to any application needing clean birdsongs.

In the future, DSSS will enable researchers to conduct more accurate bird vocalization feature tracking, trilateration for precise location estimation, and, potentially, species identification. Going forward, integrating our DSSS methodology with existing bird monitoring systems and expanding it to handle other species could greatly enhance our understanding of ecology and contribute to more effective conservation efforts.

## ACKNOWLEDGEMENTS

The Macaulay Library provided a portion of the training data. The remaining training and testing data was passively recorded by David Knobles and Preston Wilson. The City of Austin Wildland Conservation Division provided funding, permits, and staff support for the recordings taken on properties under their jurisdiction. Manual feature measurements for 4760 of the Golden-cheeked Warbler birdsongs were provided by Logan James with assistance from Brendan Allison. The source separation model training was made possible by the Texas Advanced Computing Center.

### Funding

This work was supported by Planet Texas 2050, a research grand challenge at the University of Texas at Austin; a grant from the UT Stengl-Wyer Endowment (to Timothy Keitt); and the National Science Foundation (BCS-2009669). The funders had no role in study design, data collection and analysis, decision to publish, or preparation of the manuscript.

### Grant Disclosures

The following grant information was disclosed by the authors:
University of Texas at Austin.
UT Stengl-Wyer Endowment.
National Science Foundation: BCS-2009669.

### Competing Interests

David Knobles is the CEO and president of Knobles Scientific and Analysis, LLC.

## Author Contributions

- Justin Sasek conceived and designed the experiments, performed the experiments, analyzed the data, prepared figures and/or tables, authored or reviewed drafts of the article, contributed to open-source data extraction, data cleaning, data augmenting, and model training tool on GitHub repository, and approved the final draft.
- Brendan Allison conceived and designed the experiments, performed the experiments, analyzed the data, authored or reviewed drafts of the article, and approved the final draft.
- Andrea Contina conceived and designed the experiments, authored or reviewed drafts of the article, and approved the final draft.
- David Knobles performed the experiments, authored or reviewed drafts of the article, and approved the final draft.
- Preston Wilson performed the experiments, authored or reviewed drafts of the article, and approved the final draft.
- Timothy Keitt conceived and designed the experiments, authored or reviewed drafts of the article, and approved the final draft.

## Field Study Permissions

The following information was supplied relating to field study approvals (i.e., approving body and any reference numbers):

Field permits were provided by the City of Austin Wildland Conservation Division with the approval numbers Knobles-2016, Knobles-2017, Knobles-2018, and Knobles-2019. Besides installation and servicing, there was no human involvement.

## Data Availability

The example audio files, model weights, code, and figure data are available at GitHub: https://github.com/keittlab/Birdsong-Source-Separation.

The training data, testing data, additional model weights, raw experiment data, Macaulay Library recording catalog numbers, and a snapshot of our GitHub repository are available at Zenodo: Sasek, J. (2024). Birdsong Source Separation Submission Files. Zenodo. https://doi.org/10.5281/zenodo.12791694.

The original background and GCW training data from the Macaulay Library is available at https://www.macaulaylibrary.org.

## Supplemental Information

Supplemental information for this article can be found online at http://dx.doi.org/10.7717/peerj.17854#supplemental-information.

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
