# Peer review of "Semiautomated generation of species-specific training data from large, unlabeled acoustic datasets for deep supervised birdsong isolation"

_PeerJ, doi:10.7717/peerj.17854_

## Round 0.1 · original submission · Major Revisions

As you can see, you've received two vey thorough reviews. Although expensive, I believe none is going to cause you a lot of trouble — just a lot of attention to detail! So well done and please get the edits back to me as soon as possible.

In the accompanying file, please provide details of how you respond to each and every comment.

·

Basic reporting

No comment.

Experimental design

This article suggests a method of constructing training data (by obtaining clean birdsong and background sounds from specific sites and then layering them) to facilitate the operation of existing supervised source separation models. The results show this method works well.

In line 191, the authors state they are the first to use deep supervised source separation in a bioacoustic context. Are they also the first to use supervised source separation in a bioacoustic context? If not, relevent references are necessary.

The authors sufficiently answered their current research question: "Does this method of constructing training data work?" However, they didn't introduce whether these existing supervised source separation models can operate without this method of constructing training data or with other methods. If they can, the authors need to upgrade the current research question to "Is this method better than the existing methods?"

The methods lack sufficient detail and contain vague descriptions. Line 257, "noisiest and most interesting audio", and line 273, "properly match"

Validity of the findings

No comment.

Additional comments

The current method heavily relies on site-specific datasets to obtain birdsongs and background data to construct training data. This may restrict this method to rare bird species or species of particular interest. If these species are the only target species of this method, it is fine. If not, it would be much better for the authors to write sentences in the discussion section about how to combine this method with crowed-source data to cover more species.

Reviewer 2 ·

Basic reporting

The manuscript is written in clear and professional English. The introduction provides sufficient context to situate the study within the broader field and understand its motivation. All literature references are relevant, but a few points require further support from additional citations.

The manuscript generally follows the journal's recommended structure. However, improved clarity could be achieved by better dividing the methods, results, and discussion sections. Consider separating them more distinctly.

The figures and tables are relevant to the content of the article. However, their captions lack sufficient detail for independent interpretation. Include more descriptive captions that allow readers to understand the figures and tables without relying heavily on the main text.

The source code is shared in a public repository, and the provided instructions allowed successful execution of the suggested workflows. While the main audio data is not shared, the authors provide a valid justification for this decision. To further enhance transparency, consider recommending that the authors share a small representative subset of the audio data. This would allow readers to independently follow the training workflow.

Experimental design

The introduction discusses several relevant issues, but a more focused research question would benefit readers by clarifying the study's primary objective and motivating the investigation. For example, you could suggest a refined question like: "Does leveraging locally sourced training data improve the performance of a deep supervised source separation model for birdsong isolation compared to conventional methods? Furthermore, how does this impact downstream bioacoustic tasks such as automated classification and feature extraction?"

The paper provides a thorough description of the evaluation methods, encompassing various relevant bioacoustic tasks. However, the selection criteria for evaluation data, particularly its separation from training data, needs clarification. To minimize overfitting, consider separating training and evaluation sites. While demonstrating the efficacy of source separation for downstream tasks is valuable, a robust comparison with existing source separation methods is crucial to highlight the novelty and advantages of the proposed approach. Additionally, the relatively small sample size used in the evaluation need expanding to increase the robustness of the results. The authors could consider including a statistical analysis of the results' robustness.

Validity of the findings

The paper presents interesting findings with potential relevance to the field of bioacoustics. However, some aspects could be improved to enhance the overall rigor of the study.

The authors' conclusions would benefit from stronger support derived from the study results. As previously mentioned, addressing the limited evaluation sample size and incorporating comparisons with existing methods would significantly strengthen the conclusions' foundation.

Claims regarding potential downsides of alternative approaches require stronger justification. For example, the statement about trilateration's susceptibility to background noise would be more persuasive with additional literature references or empirical evidence. Similarly, concerns about feature extraction being hindered by other source separation methods would benefit from further support.

The paper's claim that the proposed method is preferable to unsupervised source separation models needs clarification. The argument regarding the interpretability of unsupervised methods based on the lack of ground truth examples is unclear. While some unsupervised methods might focus less on the training data, this doesn't necessarily limit user control over training data selection. This issue pertains more to customization rather than interpretability. The authors should provide a clearer explanation to support this claim.

Additional comments

## Abstract

* The abstract provides a good overview but could benefit from additional details:
* Challenges: Briefly describe the specific challenges associated with evaluating performance in birdsong source separation.
* Methods: Briefly mention the utilized methods for measuring success (e.g., ornithology-inspired metrics).
* Conclusion: Expand on the abstract's conclusion to highlight the broader implications of your work for bioacoustics and ornithology.

* Line 35: Expand on "ground truth" and "sources": Consider rephrasing as "However, evaluating performance requires access to reference data (ground truth) for the isolated birdsongs (sources)." Briefly explain why this is a challenge (e.g., "often unavailable in field recordings").
* Line 38: Clarify the addressed challenges: Refer back to the challenges mentioned in the first specific comment.
* Line 40: Define "SS dataset": Change "full SS dataset" to "a complete dataset of source-separated (SS) audio recordings."
* Line 41: Change wording: Replace "a DSSS model can be trained on" with "we trained a Deep Supervised Source Separation (DSSS) model with."
* Line 42: Improve clarity: Replace "ornithology-inspired metrics" with "metrics commonly used in downstream ornithological analyses" for better understanding by a broader audience.
* Line 45: Address model testing and error type: Consider revising to "demonstrated reduced error on BirdNet accuracy, a birdsong classification model", specify error type, e.g., classification accuracy."
* Line 51: Expand conclusion: Briefly mention the implications. For example, "These findings suggest the potential of locally sourced training data to improve birdsong analysis in bioacoustic applications."

## Introduction

* Streamline the introduction by focusing on the core gap being addressed. Consider removing unnecessary details to improve readability.
* Revise the section discussing the interpretability of unsupervised models. Focus on the limitations of unsupervised approaches regarding the customizability of the training data creation process, rather than interpretability per se. Deep learning models in general, supervised or unsupervised, often lack interpretability.

* Line 66: Include a reference for the statement about acoustic richness impacting performance (e.g., a study on the limitations of automated analysis in complex audio environments).
* Line 72: Provide context before diving into definitions. Briefly discuss the role of birdsong analysis and the challenges posed by complex soundscapes.
* Line 74-79: The authors introduce the concept of "noisy birdsong" but its definition lacks clarity. It's unclear how this definition is "more realistic" than SNR measurements without a more precise explanation. The current definition seems relative to a specific analysis method. A birdsong is considered noisy if background sounds lead to misclassification. However, with Deep Learning models (often treated as "black boxes"), attributing misclassification solely to background noise is challenging. Additionally, if an alternative approach proves robust to the specific background noise in question, would the example then be considered "clean birdsong"? While SNR limitations are acknowledged (valuable discussion!), consider restructuring this paragraph. Option 1: Provide a clearer definition of "noisy birdsong" that goes beyond misclassification due to background noise. Option 2: Move this discussion to a section where it's directly relevant to the analysis methodology employed.
* Line 96-98: Clarify BirdNET training data sources to include eBird and Xeno-canto. Mention that while recordings are focal, some noisy examples are included.
* Line 110: Revise wording: Change "such an approach implicitly requires" to "current approaches typically require."
* Line 115-116: Add a reference for trilateration for animal localization.
* Line 120: Consider alternative references for limitations of trilateration with multiple sound sources. (Zhang et al., 2014 may not be the most relevant source here).
* Line 123-125: Consider providing more context to introduce "visual semantic separation." Briefly explain how this computer vision technique (classifying image pixels) is leveraged for audio source separation in spectrograms (visualizing sound frequencies). Highlight the key limitation: the assumption of a single source per time-frequency bin, which is unrealistic for audio signals where multiple sources can overlap within the same bin.
* Line 132-140 & 142-157: Consider merging and shortening these sections. Briefly discuss limitations of trilateration and other existing methods (without excessive detail) to establish the context for your proposed approach.
* Line 163: Merge with the previous paragraph as they discuss related topics.
* Line 171: Add a missing period in "similar categories For instance,".

## Materials and Methods

* Provide clear details about the evaluation data: Was a separate subset reserved for evaluation? If so, how was it chosen (e.g., random selection, stratified sampling)? If no separate subset was used, describe the evaluation methodology (e.g., k-fold cross-validation). Ideally, both background and foreground sounds for evaluation should come from a different site than the training data to avoid overfitting due to similar soundscapes.
* Define "feature labeling": Does it involve classifying birdsong types (A, B, or C)? Or does it refer to measuring acoustic features? Include details about the labeling process (e.g., manual labeling, automated methods).
* Consider increasing the number of evaluation examples. Ten examples might be a small sample size, making it difficult to draw statistically robust conclusions unless the authors demonstrate low variance in the results.
* Restructure the section to focus solely on methods. Imagine you're writing a detailed recipe for someone to replicate the experiment.
* Elaborate on how the "simulated trilateration environment" was coded. Provide more specifics about the simulation process.

## Results

* Lines 387-389: Explain "nothing similar was seen while training." Did the training process not encounter any mixed background samples, or were individual background noise samples not used?
* Line 396: Change "perverses" to "preserves." The statement "this shows that our model preserves unique variations and distortions between birdsongs" needs stronger evidence. Consider revising to "this suggests that our model may be able to preserve..." and move it to the Discussion section.
* Line 401: How much background audio was removed? Replace "most" with a quantitative measure (e.g., percentage reduction) to support the claim about background audio removal by the DSSS model.
* Line 405: Specify the interference measure used and report the numerical values of the interference for better understanding.
* Line 409-411: The high relative error reduction might be due to a low baseline error. Consider presenting results with absolute error reduction for clearer interpretation.
* Line 415: Use a more interpretable error metric like mean squared error (MSE) to show the discrepancy between original and recovered features. Increase the number of evaluation examples for robust results. Consider including a reference for acceptable error levels and provide statistical analysis to demonstrate the error falls within those limits.
* Line 436: Explain the context of "given a 22050 Hz sampling rate."

## Discussion and Conclusions

### General Comments

* The discussion currently repeats points from the Introduction and Results sections. Remove this repetition to avoid redundancy.
* Ensure the Results section focuses solely on presenting the findings, without discussion or interpretation. Move any discussion points to the dedicated Discussion section.
* Focus the discussion on broader implications of the work: What are the potential applications and real-world impacts of this research?
* Acknowledge limitations of the study: Highlight the fact that testing was limited to a single species.
* Discuss potential for broader applicability: Explore the possibility of using the method with other bird classification models (requires further testing).
* Analyze the influence of model choices: Discuss how the selection of BirdNET as the classification model might affect the results.
* Consider other relevant acoustic features: Explore how the method might impact other important features like peak frequency and duration.

## Figures and Tables

* Captions: Figures and tables should be understandable on their own, even for readers who haven't read the entire text. Provide captions that offer sufficient information about: The content being displayed (data, variables), The meaning of axes, labels, and symbols, The overall takeaway or conclusion illustrated by the figure/table.
* Tables: Include the number of samples used in the evaluation within the tables themselves. Alternatively, consider adding another column or row dedicated to statistical significance measures (e.g., p-values). This helps readers assess the reliability of the results.
* Figure 1: Group bar plots by metric type rather than other categories. This allows for easier comparison of the same metric across different scenarios. Consider using box plots instead of bar plots. Box plots effectively show the distribution of the data (including median, quartiles, and outliers) which can reveal result variance.
* Figure 2: Include images directly in the figure to visually represent: Mixed birdsong (before processing), Clean birdsong (isolated), Mixed birdsong after convolution with the forest impulse response. In the caption and figure itself, provide more details about what the reader should expect to see. Explain how the visual elements of the figure relate to the authors' claims and the overall point being made.

## Supporting Information

### General comments

* In the archived code repository, I found you include one of the dependencies "asteroid" as s subfolder within the repository. Is this intentional? Have you made any modifications to the library? If not, I would suggest removing the library from the repository and adding it as a dependency in the requirements.txt file. Otherwise, please provide more information on what modifications were made to the library and why.

---

## Round 0.2 · accepted · Accept

I've read the thorough response to the first set of reviews. I am convinced by your answers, so I see no need to send this to any other reviewers.